# Evaluation of Pyrophosphate-Driven Proton Pumps in *Saccharomyces cerevisiae* under Stress Conditions

**DOI:** 10.3390/microorganisms12030625

**Published:** 2024-03-20

**Authors:** Krishnan Sreenivas, Leon Eisentraut, Daniel P. Brink, Viktor C. Persson, Magnus Carlquist, Marie F. Gorwa-Grauslund, Ed W. J. van Niel

**Affiliations:** Division of Applied Microbiology, Department of Chemistry, Lund University, Naturvetarvägen 14, 221 00 Lund, Sweden; krishnan.sreenivas@tmb.lth.se (K.S.); leon.eisentraut@chalmers.se (L.E.); daniel.brink@tmb.lth.se (D.P.B.); magnus.carlquist@tmb.lth.se (M.C.); marie-francoise.gorwa-grauslund@tmb.lth.se (M.F.G.-G.)

**Keywords:** *Saccharomyces cerevisiae*, pH homeostasis, ATP, proton-translocating ATPase (H^+^-ATPase), proton translocating pyrophosphatase (H^+^-PPase), pHluorin, mQueen-2m, acetic acid, glucose, xylose

## Abstract

In *Saccharomyces cerevisiae*, pH homeostasis is reliant on ATP due to the use of proton-translocating ATPase (H^+^-ATPase) which constitutes a major drain within cellular ATP supply. Here, an exogenous proton-translocating pyrophosphatase (H^+^-PPase) from *Arabidopsis thaliana,* which uses inorganic pyrophosphate (PP_i_) rather than ATP, was evaluated for its effect on reducing the ATP burden. The H^+^-Ppase was localized to the vacuolar membrane or to the cell membrane, and their impact was studied under acetate stress at a low pH. Biosensors (pHluorin and mQueen-2m) were used to observe changes in intracellular pH (pH_i_) and ATP levels during growth on either glucose or xylose. A significant improvement of 35% in the growth rate at a pH of 3.7 and 6 g·L^−1^ acetic acid stress was observed in the vacuolar membrane H^+^-PPase strain compared to the parent strain. ATP levels were elevated in the same strain during anaerobic glucose and xylose fermentations. During anaerobic xylose fermentations, co-expression of pHluorin and a vacuolar membrane H^+^-PPase improved the growth characteristics by means of an improved growth rate (11.4%) and elongated logarithmic growth duration. Our study identified a potential method for improving productivity in the use of *S. cerevisiae* as a cell factory under the harsh conditions present in industry.

## 1. Introduction

The production of food supplements, such as S-Adenosylmethionine and glutathione, in yeast species such as *Saccharomyces cerevisiae* and *Candida utilis* is promising for the industrial settings due to their high flux toward these compounds [1,2]. However, recent studies have shown that the production of these compounds could be further improved by increasing ATP levels [2,3,4]. In the production of isoprenoid farnesene and polyketides in various yeasts, [5] extensive engineering aiming at reducing ATP usage was required [5,6,7]. Increased ATP levels have recently been achieved in *S. cerevisiae* through the expression of ATP-independent light-driven proton pumps [8], but its application in production strains is yet to be tested. Thus, the total ATP supply could be a bottleneck to produce fine and bulk chemicals with microbial cell factories, and attempts to improve ATP supply have already taken various approaches [9].

In *S. cerevisiae*, pH homeostasis is presumed to use a majority of the ATP produced by glycolysis [10]. This is due to the use of H^+^-ATPases (Pma1p and vacuolar ATPases) for pH homeostasis within the cell organelles, as well as the cytoplasm [10,11]. These pumps use 1 mole of ATP per translocated mole of protons [12]. Alternatively, organisms such as bacteria, archaea, and plants use inorganic pyrophosphate (PP_i_) in addition to ATP as an energy carrier [13,14]. This molecule is a byproduct of various anabolic processes, especially in DNA, RNA, and protein synthesis [15]. In yeast, however, the role of PP_i_ is not well understood and the majority of the cytosolic PP_i_ formed is hydrolyzed by Ipp1p (inorganic pyrophosphatase) producing heat [16,17,18,19]. PP_i_ has also been shown to facilitate the release of inorganic phosphate (P_i_) from vacuoles under in vitro conditions [20], but other roles are yet to be investigated. Thus, this may represent an untapped energy source, as it has been observed that PP_i_ concentrations in *S. cerevisiae* can be 10 to 1000 times higher than ATP during early growth on glucose under aerobic, as well as oxygen-limited conditions [21].

Previous studies identified conditions under which there is either an ATP limitation or ATP formation flux limitation in *S. cerevisiae* [12,22,23,24,25]. One such condition has been the growth and ethanol production from lignocellulosic biomass in the second-generation biofuel process. This substrate has numerous inhibitory compounds produced during their pretreatment steps, including acetic acid that arises from hemicellulose depolymerization [26]. The acid stress tolerance in *S. cerevisiae* is facilitated by an active export of protons via the expenditure of ATP [12,23], which is carried out by H^+^-ATPases (Pma1p and Vma3p) [27]. The upregulation of *PMA1* has also been shown to confer increased acetic acid tolerance in shake flask cultures [28]. Preadapting yeast by exposure to low levels of acetic acid also reduced its impact on ethanol productivity, but a similar genetic response was seen [29]. Another condition was the growth of engineered *S. cerevisiae* on xylose which is abundant in lignocellulosic biomass [25,30]. Under these conditions, the pH homeostasis mechanism is not well elucidated. Nevertheless, it is well understood that the proteins for pH homeostasis (H^+^-ATPases) are post-translationally regulated by glucose [10,31]. In *S. cerevisiae* strains engineered to utilize xylose via the xylose reductase/xylitol dehydrogenase (XR/XDH) pathway, a relatively higher demand for ATP was observed due to the reaction in which xylulose is phosphorylated to xylulose 5-phosphate by xylulose kinase (XK), the need to overexpress the entire pentose phosphate pathway, the high expression levels of XK, and the lack of a known negative feedback mechanism to control the ATP utilization of the enzyme [32,33].

In the current study, it was evaluated whether engineered strains of *S. cerevisiae* can utilize PP_i_ as an alternative energy carrier to improve growth and ethanol production, and under what environmental conditions this phenotype will emerge. A proton-translocating pyrophosphatase (H^+^-PPase) from *Arabidopsis thaliana* was used to assist in maintaining the intracellular pH (pH_i_) homeostasis. This protein, which is instrumental in seed germination in *A. thaliana*, has been expressed successfully in *S. cerevisiae* before and shown to confer amine fungicide resistance, as well as increased tolerance to metal stressors (Cd, Mn, and Zn); however, these studies did not focus on the impact of heterologous expression on the growth rate, productivities or their effects on cellular morphology [34,35,36,37]. Since it is known that inorganic pyrophosphate homeostasis is essential for proper yeast growth [38], an overexpression of a heterologous PP_i_-driven proton pump may impact the fermentation profiles of the yeast. To provide additional real-time measurements, the pH_i_ and ATP levels were studied using the pHluorin and mQueen biosensors [39,40,41]. The present study expands on these earlier studies by examining the impact of the *A*. *thaliana* H^+^-PPase on the pH_i_ homeostasis of engineered *S. cerevisiae* strains, especially during stress conditions that exist within industrial settings: (a) growth at a low pH in the presence of acetic acid and (b) xylose fermentation, using glucose fermentation as the control. The H^+^-PPase was targeted toward either the cell membrane or the vacuolar membrane, and the impact of its localization on cell morphology, growth, physiology, pH_i_ and ATP levels, was evaluated. The relatively small GFP-based biosensors used for pH_i_ and ATP usually have a minimal impact on growth, but manifested varying degrees of impact on growth depending on the severity of the environmental conditions.

## 2. Materials and Methods

### 2.1. Strains and Maintenance

All yeast strains used in this study are mentioned in Table 1. *Escherichia coli* NEB5ɑ (New England Biolabs, Ipswich, MA, USA) made chemically competent was used as the general cloning host for plasmid construction and maintenance [42]. All organisms and strains were stored in 25% glycerol at −80 °C. 

The *E. coli* strains were grown and maintained in a Luria broth (LB) [44] supplemented with ampicillin (100 µg·mL^−1^) and agar (15 g·L^−1^) whenever necessary. All *E. coli* cultivations were carried out at 37 °C for 16 to 18 h. 

The yeast strains were revived and maintained in YPD (yeast extract 10 g·L^−1^, peptone 20 g·L^−1^, D-glucose 20 g·L^−1^) supplemented with geneticin (200 µg·mL^−1^) and nourseothricin (100 µg·mL^−1^) for CRISPR-Cas9 modifications. All yeast cultivations were carried out at 30 °C. 

### 2.2. Plasmid Construction

The plasmids used in this study are listed in Table 2. pUC57-VP-Suc2PSP-AVP1 contained a yeast codon-optimized vacuolar H^+^-PPase sequence (*AVP1*) from *Arabidopsis thaliana* fused to signal peptides consisting of both the N-terminal domain of *Trypanosoma cruzi* H^+^-PPase, followed by the N-terminal domain of the endogenous *S. cerevisiae* invertase (the first 25 amino acids from Suc2p), as mentioned in [34], were synthesized by GenScript (Piscataway, NJ, United States). Restriction enzymes, Phusion High-Fidelity DNA polymerase, DreamTaq polymerase, and T4 DNA ligase (5 U·µL^−1^) were obtained from Thermo Fisher Scientfic (Waltham, MA, United States). Plasmid extraction, gel extraction and PCR purification were carried out using GeneJET extraction kits (Thermo Fisher Scientific) according to the manufacturer’s protocols. Primers were obtained from Eurofins Genomics (Ebersberg, Germany). The plasmid pTMB_KS_042 was constructed by amplifying pUC57-VP-Suc2PSP-AVP1 with the primer pair 88—Tc_VP_R and 88r—AVP1_FW, and was blunt-end-ligated. The plasmid pTMB_KS_043 was constructed by amplifying pUC57-VP-Suc2PSP-AVP1 with the primers 89—Suc2SP_F and 90—KS_Tef1p_Rev, and was blunt-end-ligated. The cassettes of pTMB_KS_044 and pTMB_KS_045 were obtained by cloning using BamH1 and KpnI restriction sites.

The fluorescent sensor cassette for pH_i_ was constructed by amplifying the pHluorin gene from pYES-pACT1-pHluorin using 84r—phlu_BamH_F and 85—phlu_EcoRI_R and cloned into p426GPD using BamHI and EcoRI sites to obtain pTMB_KS_036. The fluorescent sensor cassette for monitoring intracellular ATP levels was constructed by traditional cloning of the Queen2m gene from pRSET-QUE2m into p426GPD using BamHI and EcoRI sites to attain pTMB_KS_038. 

The plasmids for integration of the sensor cassettes into the X-4 intergenic site designed by Jessop-Fabre et al. (pTMB_KS_040 and pTMB_KS_041) targeted by pCFB3042 [46] were assembled from pTMB_KS_036 and pTMB_KS_038 by amplification with NM_CYC7tSdaI_RV_R and NM_GPDpSac1_FW_F, and restriction cloning into pCFB3035 using SdaI and SacI sites. All plasmids constructed were confirmed using Sanger sequencing from Eurofins Genomics (Ebersberg, Germany), using the primers listed in Appendix A.

### 2.3. Yeast Strain Engineering

The different strains of *S. cerevisiae* were obtained using the CRISPR/Cas9 system [46]. High-efficiency competent yeast cells were obtained by using a modified version of the lithium acetate method [47] that used 10% (*v*/*v*) DMSO prior to the heat shock treatment. TMB 3504 containing pCFB2312 was transformed with pCFB2904 in addition to NotI linearized pTMB_KS_044 or pTMB_KS_045 to acquire the vacuolar membrane H^+^-PPase strain (TMB_KS_S02) and the cell membrane H^+^-PPase strain (TMB_KS_S03), respectively.

Strains TMB 3504, TMB_KS_S02, and TMB_KS_S03 were subjected to subsequent transformation using pCFB2312, pCFB3035, and NotI linearized pTMB_KS_040, and pTMB_KS_041 to obtain strains TMB_KS_S04 to TMB_KS_S09. The verification of the transformants was achieved by colony PCR using primers mentioned in Appendix A.

### 2.4. Microtiter Plate Cultures

Single colonies of the different strains were inoculated into 5 mL of a Verduyn mineral medium [48] at pH 5.0 containing 20 g·L^−1^ of glucose. The culture was incubated for 24 h to 26 h at 30 °C and 180 rpm, washed with sterile H_2_O and then used to inoculate a 96-well microtiter plate containing a Verduyn mineral medium at pH 5.0 or pH 3.7 with 20 g·L^−1^ of glucose and 0, 3, or 6 g·L^−1^ of acetic acid to an optical density (OD) of approximately 0.1. The pH of the mineral medium was adjusted to 3.7 using 3 M KOH after adding glacial acetic acid or with 2 M H_2_SO_4_ when no acetic acid was added. The OD was measured at 620 nm every 2 h in an automated spectrophotometer (Multiscan Ascent, Thermo Electron Corporation, Waltham, MA, USA) maintained at 30 °C with background shaking for 2 min at 8 min intervals. A total of three technical replicates (3 wells) per biological triplicate (3 independent clones) were performed for each condition. At the end of the growth experiment, cell count was carried out by using a MACSQuant VYB with an uptake volume of 15 µL culture and a flow rate of 25 µL·min^−1^ to get the correlation of the cell number to OD_620_.

The maximum growth rates were estimated by using the logistic model in Growthcurver [49] on R v4.2.2. The growth rate was determined using all the data in the technical triplicates for each biological triplicate. The growth rates and the cell counts were subjected to an ANOVA followed by a Tukey’s post hoc test whenever applicable.

### 2.5. Growth Studies in Bioreactors (or Growth Studies or Cultivations)

A 3 L bioreactor (Applikon, Schiedam, The Netherlands), equipped with an ADI 1025 Bio-Console and an ADI 1010 Bio-Controller, was used for culturing 2 different biological clones of the yeast strains at a working volume of 1 L. The pH of the medium was maintained at 5.0 ± 0.1 by automatic titration of 3 M KOH. The bioreactor was kept at 30 °C using an electric heat blanket and the stirring was set at 400 rpm.

For fermentations on glucose, the pre-culture was prepared by inoculating a single colony from a YPD plate into 30 mL of Verduyn mineral medium with glucose (20 g·L^−1^) and grown aerobically in baffled shake flasks for 24 to 26 h. The bioreactor was sparged continuously with N_2_ at 400 mL·min^−1^, and inoculations of 0.01 g CDW·L^−1^ were carried out after 2 h of sparging to ensure anaerobic conditions. 

The pre-cultures for xylose fermentations were started by inoculating a single colony into 5 mL of YPD and incubated overnight. The overnight culture was washed, and the resuspended cells were inoculated in a sealed flask to an OD_620_ of 0.3 in a 50 mL Verduyn mineral medium with xylose (50 g·L^−1^) and incubated under oxygen-limited conditions in 250 mL serum vials with a 23 gauge 1″ needle connected to a sterile syringe filled with cotton as a gas vent and incubated for 36 to 48 h. The bioreactor was sparged at 250 mL N_2_·min^−1^ for 2 h and subsequently, the bioreactor was sealed after inoculation with 0.03 gCDW·L^−1^. This condition was sustained until the biomass reached 0.06 gCDW·L^−1^, after which N_2_ sparging was reinitiated. Up to 5 mL of samples were frequently taken for OD, flow cytometry and HPLC analysis. Of the samples, 25 mL was taken for cell dry weight measurements at 3 time points during logarithmic growth.

### 2.6. Analytical Methods

Continuous CO_2_ analysis in the effluent gas was performed using either a BlueVary (BlueSens gas sensor GmbH, Herten, Germany) or a Tandem PRO Gas analyzer (Magellan Instruments Ltd., Limpenhoe, UK). The trends for CO_2_ production were used to determine the end of the batch fermentations. The Tandem PRO Gas analyzers logged data at 10 s intervals, whereas the BlueVary gas analyzer had 5 s intervals.

Organic acid detection was performed using a HPLC (Waters, Milford, MA, USA) equipped with an Aminex HPX-87H ion exchange column being maintained at 60 °C with 5 mM H_2_SO_4_ as an eluent pumped at 0.6 mL·min^−1^. Detection was performed using a refractive index detector (Shimadzu, Tokyo, Japan). Optical density measurements were performed using an Ultraspec 2100 pro spectrophotometer (Amersham Biosciences, Little Chalfont, UK) at 620 nm. Cell dry weight measurements were obtained by filtering 10 mL of culture through 47 mm Supour^TM^ 450 membrane disc filters (Pall Life Sciences, Port Washington, NY, USA) [50]. Phase contrast microscopy was carried out during the logarithmic growth phase using a Leica DM750 Microscope equipped with a ICC50W camera module (Leica Microsystems, Wetzlar, Germany).

### 2.7. Flow Cytometry

All flow cytometry analyses for growth studies in bioreactors were carried out with a MACSQuant^®^ VYB flow cytometer (Miltenyi Biotec, Bergisch Gladbach, Germany) applying a flow rate of 50 µL·min^−1^ and a threshold of 0.83 on the forward scatter. The cell suspensions were diluted to an OD of 0.1 to 0.2 using PBS with 1.32 µg·mL^−1^ of propidium iodide [51]. The measurement of the fluorescence for the biosensors was started within 10 min of sampling. For obtaining standard curves for the pHluorin sensor, samples taken from the bioreactor were spun down and resuspended in a PBS buffer (pH 7.4) with 0.04 mM Digitonin and incubated with shaking at 30 °C for 15 min. The cells were then spun down and resuspended in 0.2 M phosphate buffer at pH 5.7, 6.4, 7.0, and 8.0 [39]. The zero value for the mQueen ATP sensor was obtained by resuspending the sample from the bioreactor in a Verduyn medium supplemented with 0.5 g·L^−1^ 2-deoxy-D-glucose and incubated at 30 °C for 1 h [41,52]. 

The measurements of the pH_i_ with pHrodo^®^ green required a staining period of 30 min in an LCIS buffer according to the manufacturer recommendations. The protocol for pH_i_ using this dye was performed on one of the xylose fermentations conducted in bioreactors on the strains without biosensors in technical replicates and it constituted the resuspension of a cell pellet to an OD of 0.1 to 0.2 in an LCIS buffer (pH 7.4) with 4 µM of pHrodo^®^ green and incubation at 30 °C for 30 min followed by centrifugation and resuspension in a PBS buffer (pH 7.0). The flow cytometry used the same instrument setting mentioned above. For the calibration curve for pH estimation using pHrodo^®^ green, a larger volume of cells was stained and then incubated at 70 °C with 0.04 mM digitonin for 15 min and then split into 4 tubes and resuspended in citrate–phosphate buffers at pH 4.6, 5.6, 6.6, and 7.6.

The obtained flow cytometry data for the biosensor strains were processed in MACSQuantify^TM^ (version 2.13.3). The data were subjected to a polygonal exclusion (not) gate on the area plots of B1 vs. B2 (525/50 nm vs. 593/50 nm) to select for non-permeable fluorescent cells. These fluorescent cells were then subjected to an ellipsoid gate on the area plots of forward scatter (FSC-A) and side scatter (SSC-A) to eliminate instrument noise. From the remaining events, the means of V2-A (525/50 nm, emission from the violet laser) and the B1-A (525/50 nm, emission from the blue laser) were obtained. These values were used to obtain the emission ratios (405/488 nm) from excitation at 405 and 488 nm. The obtained ratio values were used in a linear equation obtained from the ratio values of the fluorescence emission of the standard curves to obtain pH_i_ values. For the intracellular ATP levels, the ratio of V2 over B1 from the cells resuspended in 0.5 g·L^−1^ 2-deoxy-D-glucose was taken as the zero value for each strain and the relative ratio of the samples was used as an estimate of ATP levels.

The flow cytometry data for the pHrodo^®^ staining were processed in FlowJo (Version 10.10.0, Benton Dickinson & company (FlowJo LLC), Ashland, OR, USA). The data from the stained cells were subjected to a gate on the area plot of B1 vs. B2 (525/50 nm vs. 593/50 nm) to remove the propidium iodide-stained cells. The geometric mean of B1-A for the live cells was used for pH_i_ estimation. The samples for the standard curve were not subjected to propidium iodide staining but instead permeabilized using digitonin, and thus these cells were subjected to a polygonal gate on the area plots of B1 vs. side scatter (525/50 nm vs. 561/4 nm) to differentiate live and permeable cells. The geometric mean of the B1-A for the permeable cells was used to make the standard curve (Appendix A).

### 2.8. Calculations

The estimated total cell counts for the microtiter plate experiments were obtained using the total number of events per µL (quantified by flow cytometer) multiplied with the total volume (200 µL).

The correlation between cell dry weight (*CDW*) and *OD_620_* was calculated to estimate the cell dry weight of each sample (Equation (1)).
(1)$$CDW\;\left(g\cdot L^{-1}\right)=a\left[{OD}_{620}\right]+0$$
where *a* is the slope of the linear regression and 0 is the linear regression constant forced to 0. The values of *a* varied between strains and conditions.

The specific production or consumption rate (*q*) at *µ_Max_* was determined by multiplying the slope between the substrate or product, and the corresponding *CDW*, with a maximum growth rate (*µ*) (Equation (2)).
(2)$$q\;\left(g_{product\;or\;substrate}\cdot g_{CDW}^{-1}\cdot h^{-1}\right)=\frac{\Delta \;Product\;or\;Substrate}{\Delta \;CDW}\times {µ}_{Max}$$
where *q* is the specific consumption or production rate, Δ*Substrate* refers to either glucose or xylose and Δ*Product* refers to acetate, xylitol, glycerol or ethanol during logarithmic growth. Δ*CDW* is the difference in cell dry weight over the same period. The period of logarithmic growth was identified by plotting the natural log of the CDW over time. 

The volumetric consumption or production rate (*Q*) was obtained by multiplying the Δ*product* or Δ*substrate* during logarithmic growth with the maximum growth rate (µ) (Equation (3)).
(3)$$Q\;\left(g_{product\;or\;substrate}\cdot L^{-1}\cdot h^{-1}\right)=\frac{\Delta \;Substrate\;or\;Product}{Volume\;(1\;L)}\times {µ}_{Max}$$
where *Q* is the volumetric consumption or production rate, Δ*Product* is acetate, ethanol, glycerol or xylitol, and Δ*Substrate* is glucose or xylose. The volumetric production rate was calculated over the same period as specific production or consumption rate.

The total yield of products was calculated using Δ*Product* divided by Δ*Substrate* over the entire fermentation duration (Equation (4)).
(4)$$Y\;\left(g_{Product}\cdot g_{Substrate}^{-1}\right)=\frac{\Delta \;Product}{\Delta \;Substrate}$$
where *Y* is the yield calculated over the entire fermentation, Δ*Product* is the amount of acetate, ethanol, glycerol or xylitol produced, and Δ*Substrate* is the amount of glucose or xylose consumed.

## 3. Results and Discussions

### 3.1. Vacuolar Membrane H^+^-PPase Improved Growth Rates at a Low pH and Acetic Acid Stress

Since pH homeostasis is known to use significant amounts of ATP during a series of acetic acid stress [10,53], the impact of the addition of H^+^-PPase on the growth of *S. cerevisiae* was tested under these stress conditions. This was to see if PP_i_ could compensate for the energy drain caused by acetic acid, thereby increasing the growth rate. *A. thaliana* H^+^-PPase was targeted either to the cytosolic or the vacuolar membrane, and the strains were evaluated for growth in microtiter plates in 20 g·L^−1^ of glucose supplemented with 0, 3, and 6 g·L^−1^ of acetic acid at pH 5.0 and 3.7 under oxygen-limited conditions.

There was a significant increase in growth rate upon the additional expression of the H^+^-PPase in the vacuolar membrane (strain TMB_KS_S02) as compared to the control strain (TMB 3504) in mineral media at pH 3.7 and 6 g·L^−1^ of acetic acid (Figure 1A). This supported our hypothesized decrease in ATP burden from using PP_i_ as an alternative energy carrier. These growth rates were obtained by fitting a logistic model through the optical density values of all three technical replicates of a biological replicate (Appendix A). The cell membrane H^+^-PPase strain (TMB_KS_S03) showed significantly lowered growth rates in all conditions and failed to grow at pH 3.7 and 6 g·L^−1^ acetic acid. Tukey’s post hoc test was performed for all conditions as the ANOVA showed significant *p*-values (pH 5 (0 g·L^−1^, *p* = 0.022; 3 g·L^−1^, *p* = 0.015; 6 g·L^−1^, *p* = 0.003), pH 3.7 (0 g·L^−1^, *p* = 0.021; 3 g·L^−1^, *p* = 0.008; 6 g·L^−1^, *p* = 0.0002)).

The parent strain showed high variability in the lag phase between the biological replicates (Figure 1B), ranging from 28 h to over 50 h at pH 3.7 and 6 g·L^−1^ of acetic acid. Although it is known that adapting the preculture reduces the lag phase variations under these conditions [29,54], we still wanted to investigate if the H^+^-PPase could complement the pre-adaptation step. When targeted to the vacuolar membrane (TMB_KS_S02), the H^+^-PPase reduced the lag phase to below 28 h in all biological replicates (Figure 1B). The variations in the lag phase at lower concentrations of acetic acid at pH 3.7 were not as well pronounced (Appendix A). It was also observed that the vacuolar and cell membrane H^+^-PPase strains (TMB_KS_S02 and TMB_KS_S03) displayed a larger variation in the total estimated cell count determined using flow cytometry between biological replicates (Appendix A).

The direction of the reaction catalyzed by the *A. thaliana* H^+^-PPase used in this study was determined by the proton and inorganic pyrophosphate (PP_i_) concentration [55]. These concentrations are known to vary naturally during fermentation [21]. Upon expressing the H^+^-PPase, we hypothesized that at different growth phases, the H^+^-PPase and ATPase are counteracting each other, and at other growth phases, they are acting synergistically. This would have to be evaluated further and was not within the scope of the present study. The reduction in growth rate in the strain expressing the H^+^-PPase at the cell membrane may also be attributed to this effect. In the cell membrane-targeted H^+^-PPase, the direct exposure of the pump to the acidic extracellular environment most likely reversed the enzyme direction, as it is also dependent on the proton and PP_i_ concentrations [55]. However, as the external pH was not maintained in the microtiter plate set-up, an evaluation of the pH_i_ could not be performed.

### 3.2. Expression of H^+^-PPase Led to an Acidified Cytoplasm during Glucose Fermentations

The expression of the H^+^-PPase at the vacuole improved the growth rates and reduced the lag phase at a low pH; however, at pH 5.0, there was very little variation in these parameters. To evaluate whether this expression influenced the pH_i_ and/or cytosolic ATP under two common fermentation conditions, we integrated a pH-sensitive fluorescent biosensor (pHluorin) and an ATP binding fluorescent biosensor (QUEEN-2m) into the genome. The effect of heterologous expression on fermentation profiles and physiological changes were monitored in anaerobic batch cultures on mineral media with glucose under defined conditions. 

Regardless of which combination of biosensors were expressed with the vacuolar or cell membrane H^+^-PPase, few differences were observed in growth rates (Figure 2A), specific production rates (Appendix A), volumetric production rates (Appendix A), yields (Appendix A), or metabolite profiles (Appendix A). These results are in-line with previous literature reporting that upon heterologous protein expression, yeast strains can compensate for the ATP drain caused by the protein expression. The cell accomplishes this by altering the ratio of metabolites formed or changing fluxes toward other intracellular metabolites [56,57,58,59].

In *S. cerevisiae*, pH_i_ homeostasis is mediated by proton-pumping ATPases distributed over the organelles and the cell membrane [10,11,60]. This machinery is regulated by glucose at both the transcriptional and the protein levels [31,61] and has a major impact on the ATP concentration. Thus, the introduction of H^+^-PPases, which has been successively carried out before [35,36,37], may influence the ATP concentration even further. Therefore, the physiological effect of these heterologous pumps was evaluated using fluorescent biosensors for monitoring the pH_i_ and ATP levels.

During glucose fermentation, a slight acidification of the intracellular pH was observed upon the introduction of the H^+^-PPases targeted to either the vacuolar or the cytosolic membrane (Figure 2B) compared to the control strain, TMB_KS_S04, that expressed pHlourin but no exogenous H^+^-PPase. The pH_i_ stabilized at about 7.2 for the parent strain carrying pHluorin (TMB 3504) during logarithmic growth. The pH_i_ stabilized at 7.0 and 6.9 for the vacuolar membrane H^+^-PPase strain with pHluorin (TMB_KS_S06) and the cell membrane H^+^-PPase strain with pHluorin (TMB_KS_S08), respectively. Statistical testing was not performed as the data are from biological duplicates. The pH_i_ for the parent strain with pHluorin (TMB_KS_S04) was 7.2, which is consistent with to that obtained in previous literature using alternative methodologies [11,39]. The acidification of the cytoplasm can possibly be explained by the ability of the H^+^-PPase to function as a PP_i_ synthase when high levels of ATP and P_i_ are available [55]. Any effects of the suspected increase in PP_i_ concentrations in the cytoplasm were, however, not immediately apparent and require further investigation.

The ATP levels (Figure 2C) are represented in terms of a fluorescence emission ratio obtained at 525/50 nm. An increased ratiometric value compared to the ATP depletion condition (Dashed grey line, Figure 2C) corresponds to an increase in cellular ATP levels. The ATP depletion condition was established by resuspending the cells in minimal media with 2-deoxy-D-glucose as substrate for 1 h [41,52,62,63]. The ATP levels for the vacuolar membrane H^+^-PPase strain with QUEEN-2m (TMB_KS_S07) stabilized at a higher value. This could be due to an alteration in pH homeostasis by the introduced H^+^-PPase that can operate in both directions as stated above. The measurement of the ATP levels was at discrete time points; hence an overall real time measurement would be of interest to understand the interplay between levels of ATP, P_i_, pH_cytosolic_ and pH_Vacoular_.

The lack of variation in growth rates due to an overexpression of proteins is consistent with other reports [41,52]. Additionally, it has been hypothesized that actively growing cells have sufficient pools of intracellular metabolites to accommodate for changing requirements such as the heterologous expression of proteins [64]. The reduction in pH_i_ and the increased ATP levels seen in the derivatives of TMB_KS_S02 require further investigation. Furthermore, the expression levels and the activity of the native pH homeostasis machinery, as well as the heterologous expressed proton pumps are of interest but are out of the scope of this study.

### 3.3. Vacuolar Membrane H^+^-PPase Improved Xylose Fermentation When pHluorin Is Co-Expressed

The industrial production of ethanol from lignocellulose is usually performed with *S. cerevisiae* due to its long-term industrial use, high ethanol and stress tolerance, and high ethanol yield from hexose sugars [30]. Acetic acid is commonly formed during the pre-treatment of lignocellulose biomass prior to fermentation, and thus, the improved performance by the strain expressing the vacuolar membrane H^+^-PPase is promising for valorization of lignocellulose streams. However, growth on xylose, an abundant pentose sugar in many types of lignocellulosic biomass, has required engineering of *S. cerevisiae* through introducing functional xylose-assimilation pathways [30]. In the strains with the XR/XDH pathway, limited ATP formation fluxes under anaerobic conditions have been discussed as a possible reason for the poor xylose utilization [24,25]. Therefore, xylose fermentation provides another relevant stress condition for evaluating whether the presence of the vacuolar membrane H^+^-PPase can be valuable. 

Overall, strains expressing either a H^+^-PPase (cell membrane or vacuolar), a biosensor (pHlourin or QUEEN-2m), or a combination of the two, displayed a reduction in growth rate when grown on xylose (Figure 3). The maximum growth rate was reduced to 85% and 76% of the parent strain growth rate upon the expression of either the vacuolar or cell membrane H^+^-PPase (TMB_KS_S02 and TMB_KS_S03), respectively. Similarly, the introduction of pHluorin and QUEEN-2m (TMB_KS_S04 and TMB_KS_S05) in the parent strain resulted in a reduced growth rate of 77% and 72%, respectively. There was also a reduction in the total biomass production. These observations indicated that the expression of the biosensors resulted in a metabolic burden for this parent strain during the applied cultivation conditions. The reduction in the growth rate, due to the expression of the H^+^-PPase, is likely caused by a combination of two factors. One is a high expression, due to the *TEF1* promoter [33,65], and the other is the ability of the H^+^-PPase to function in the reverse direction [55]. Regarding the first factor, in previous studies, GFP under a weak promotor was expressed in similar *S. cerevisiae* strains that did not lead to an impact on growth rate most probably because it allowed for low expression [66,67]. In the current study, using promoters that allowed for higher expression levels of the biosensors was required to compensate for the lower fluorescence levels of the biosensor proteins under anaerobic conditions. This elevated expression can be compared to how the expression of heterologous proteins for production purposes typically affects growth negatively [56,58,59].

No substantial synergistic effect was seen for most combinations of heterologous proteins, but the co-expression of the ATP biosensor (QUEEN-2m) did not affect the vacuolar H^+^-PPase strain (TMB_KS_S07) in terms of the growth rate and total biomass produced as opposed to the parent strain with the ATP biosensor (TMB_KS_S05). 

The expression of pHluorin, in addition to the cell membrane H^+^-PPase strain (TMB_KS_S08), further reduced the growth rate to 69% compared to the parent strain (TMB 3504), although the total biomass formation remained unaffected.

Remarkably, the maximum growth rate of the vacuolar H^+^-PPase strain with pHluorin (TMB_KS_S06) increased by 11.4% compared to the parent strain (TMB 3504). The strain also grew exponentially for a longer duration, reducing the overall fermentation period by about 35 h, thereby consuming 97% of the xylose (Figure 4B, Appendix A). The growth profile of this strain is similar to the growth profiles seen on glucose fermentations (Appendix A). In contrast, the parent strain (TMB 3504), the vacuolar membrane H^+^-PPase strain (TMB_KS_S02), and the cell membrane H^+^-PPase strain (TMB_KS_S03) grew in a linear fashion that took about 35 to 48 h longer to consume the same amount of xylose (Figure 4A, Appendix A). The increased duration of exponential growth also led to the consumption of approximately 90% of the xylose by 87 h for the vacuolar membrane H^+^-PPase strain compared to around 70% for the parent strain (TMB 3504) during the same time frame (Appendix A). The result can be hypothesized to be due to the energy demand caused by the production of pHluorin, redirecting the ATP from pH homeostasis towards heterologous protein production. This in turn forced the H^+^-PPase to pump protons into the vacuole and elevating the ATP burden for pH homeostasis. There are likely other factors that may also contribute to the observed increase in growth rate and reduction in fermentation duration, but these remain elusive without additional testing of intracellular metabolites. Thus, conclusive evidence of this phenomenon requires further investigation which is not within the scope of this preliminary study.

The volumetric production and consumption rates and specific productivities, (Figure 5) were calculated during logarithmic growth. The average specific productivity of ethanol was elevated by 9% for the vacuolar membrane H^+^-PPase strain carrying the pHluorin biosensor (TMB_KS_S06) (Figure 5A). 

The vacuolar membrane H^+^-PPase strain with pHluorin (TMB_KS_S06) showed a 25% increase in the mean volumetric ethanol productivity, 15% increase in the mean volumetric xylitol productivity, and a 66% increase in the mean volumetric xylose consumption rates compared to the parent strain (TMB 3504). Thus, by adding this protein expression burden apparently, a mechanism is triggered, thereby improving the performance of the cell*,* which was absent in the strain with only the vacuolar membrane H^+^-PPase (TMB_KS_S02). This requires additional testing with an expanded array of various energy drains in the form of heterologous proteins or nutrient-limiting conditions and is of future interest. The additional expression of mQueen in the vacuolar membrane H^+^-Ppase (TMB_KS_S07) strain did not give the same response as the vacuolar membrane H^+^-Ppase strain with pHluorin (TMB_KS_S06), but it is noted that the strain completed xylose utilization 32 h earlier than the strain with only the vacuolar membrane H^+^-Ppase (TMB_KS_S02) (Appendix A). All the strains, except the vacuolar membrane H^+^-PPase with pHluorin, consumed the majority of xylose during non-logarithmic growth (after 48 h) (Appendix A). This could be due to the additional burden caused by QUEEN-2m as it is a larger protein than pHluorin (380 AA vs. 238 AA) thus requiring more ATP. Furthermore, through binding the ATP, the biosensor keeps it out of the catalytic pool even though ATP is not converted [40,41].

Various stresses have been observed to affect the profiles of fermentation products, for instance, a redox imbalance stress leading to an increase in glycerol production [68]. However, although the production and consumption rates were affected by some combinations of the expression of the biosensors and H^+^-PPases, the product yields calculated over the entire fermentation period of the various strains did not differ considerably (Figure 5C).

The carbon dioxide production profiles for the strains with pHluorin were normalized to the strains without pHluorin to better examine the impact of the biosensor as it affected growth and fermentation duration. The parent strain engineered to express pHluorin (TMB_KS_S02) displayed a peak plateau of CO_2_ production for a period of about 50 h compared to the parent strain (TMB 3504) (Figure 6A). This corresponded to the period of linear growth seen in the parent strain with pHluorin (TMB_KS_S04), (Appendix A). It further showed that the CO_2_ production of the parent strain (TMB 3504) has an elongated decline phase as opposed to the very sharp decline seen on glucose fermentations (Appendix A). A similar profile was also seen for the cell membrane H^+^-PPase strain (TMB_KS_S03) and the cell membrane H^+^-PPase strain with pHluorin (TMB_KS_S08).

The vacuolar membrane H^+^-PPase strain with pHluorin (TMB_KS_S06) showed a normal batch CO_2_ profile with a sharper decline in CO_2_ production after reaching peak metabolic activity with almost no CO_2_ produced after 115 h (Figure 6B). This behavior was not seen in the parent strain (TMB 3504) or the vacuolar membrane H^+^-PPase strain (TMB_KS_S02) on xylose fermentations, which, again, indicates that the fermentation performance was enhanced by the co-expression of the vacuolar membrane H^+^-PPase and pHlourin.

The acidification of the cytosol on xylose in native xylose-utilizing yeast strains, such as *Candida tropicalis* and *Pichia stipitis,* as well as wild-type *S. cerevisiae* which cannot use xylose, has been well established [52,69,70]. Furthermore, there is evidence of pH_i_ being a signaling mechanism for sugar sensing [60,61,71]. The manipulation of pH_i_ and its impact on cell growth is of particular interest for the parent strain (TMB 3504) since it is engineered to take up xylose, but the mechanism for pH_i_ maintenance during xylose fermentation remains unknown. 

The expression of pHluorin changed the growth profiles of the strains on xylose. Additionally, the pH_i_ measurements measured using pHluorin could be trusted only until 45 h into the fermentation since the background GFP autofluorescence in the emission at B1 (525/50), due to excitation at 488 nm, increased and interfered with the calibration curves. It is seen that the vacuolar membrane H^+^-PPase strain with pHluorin (TMB_KS_S06) maintained the pH_i_ around 7.2, whereas the pH_i_ for the parent strain with pHluorin (TMB_KS_S04) and the cell membrane H^+^-PPase strain with pHluorin (TMB_KS_S08) had cytosolic alkalinization leading to a pH_i_ around 7.5 (Figure 7A). Statistical testing was not performed as the data are from biological duplicates. Previous literature also reported similar results using a similar biosensor on a co-fermentation of glucose and xylose for xylonate production [72].

Since the growth profiles of the strains changed due to the expression of pHluorin, a second method for pH_i_ measurements was performed using pHrodo^®^ green (Thermo Fisher Scientific, Waltham, MA, USA). This method indicated that there may be a change in the pH_i_ caused by the presence of pHluorin (Figure 7A,B), but the pH_i_ values obtained using pHrodo^®^ green were measured at about 45 min after sampling due to the required staining period and this likely affected the readings. It is known from live cell imaging that pH_i_ changes are observable in the order of a few minutes in metabolically active cells [41]. Nevertheless, the pHrodo^®^ green method indicated that the pH_i_ was maintained at near-normal conditions. Alternative non-invasive methods to study pH_i_ are yet to be tested and require further investigation for verification of pH_i_ in the strains without the biosensors.

The intracellular ATP levels on xylose (Figure 7C) were higher in the vacuolar membrane H^+^-PPase strain with mQueen (TMB_KS_S07) than that of the parent strain with mQueen (TMB_KS_S05), (Figure 7C). This did not correspond to an increased growth rate or an elongated exponential growth phase. This could be due to the burden of expression for mQueen-2m being higher than pHluorin or the ability of mQueen-2m to bind and dissociate with ATP, thereby removing ATP from the catalytic pool. The variations in growth characteristics between the strains with and without biosensors most likely implies that the energy metabolism was different in the various strains. The measurements of intracellular ATP were also affected by the increasing auto-fluorescence after 47 h. Various literature identified that the ATP levels vary based on the quantity and type of carbon source supplied (e.g., glucose, glycerol, galactose) [73,74]. This is compounded by the evidence for the energy and redox ratio controlling glycolytic flux [75,76]. This, coupled with the reduced ATP formation flux, lack of negative feedback for xylose uptake, and other factors (sugar uptake, sugar signaling, etc.) could lead to a very complex metabolic phenomenon [24,25,30,32]. Thus, to further elucidate the mechanisms involved in the observed increase in ATP levels will require an investigation of the total energy state of the cells.

From the various results it is seen that the heterologous expression of pHluorin in addition to the vacuolar membrane H^+^-PPase strain led to an improvement in the growth rate (11.4%), volumetric ethanol productivity (25%), the total duration of logarithmic growth, and reduced fermentation duration (20%). However, it is noted that the protein expression burden usually has a negative impact on growth rate [64,77,78]; therefore, additional tests for the type and/or quantity of burden, or additional factors that are as of yet unknown, are required. Other reports also show that the adenylate energy charge (ATP/AMP) of the cell is strictly regulated and sustained even during xylose fermentation, but only the guanylate energy charge shows variations [79]. This requires further investigation as neither the total ADP and AMP nor the guanylate charge were measured but the total ATP concentration is elevated in the vacuolar H^+^-PPase strain. These determinations are called for, as a comprehensive study of the pH homeostasis mechanism on yeast engineered for xylose utilization has not been conducted. However, the proteins involved in this function constitute a large portion of the proteome and is considered to be the highest energy utilizing mechanism [11,23,53,80]. This study presents potential beneficial effects for vacuolar H^+^-PPase expression on valorization of lignocellulosic biomass to value-added products as well as heterologous protein production.

### 3.4. Cell Morphology Is Influenced by Biosensor Expression on Xylose

Regular microscopic evaluations of the cultures in the bioreactors growing on glucose or xylose were carried out during early logarithmic growth and the stationary phase. It demonstrated that in the xylose fermentations, different degrees of aggregation were observed in the parent strains with biosensors (TMB_KS_S04 and TMB_KS_S05) and the cell membrane H^+^-PPase strains with biosensors (TMB_KS_S08) (Appendix A). This was completely absent in the glucose-growing cultures, as well as the strain without biosensors grown on xylose. Of note is that no aggregation was seen on the vacuolar membrane H^+^-PPase strains and its derivatives with biosensors (TMB_KS_S02, TMB_KS_S06 and TMB_KS_S07). 

At this stage, it remains unknown how far this diversity in morphology has influenced the OD measurements, and therefore, also the dry weights that were estimated from the OD values. The difference in calibration slopes of OD versus dry weight (Section 2.8) might reflect this influence. However, the carbon and redox balances did not show deviations due to the aggregation (Appendix A).

The aggregating behavior raises new questions that merit specific attention, such as what initiates the aggregation, what type of aggregation it is, and if the expression of the biosensors is involved in this process.

## 4. Conclusions

*S. cerevisiae* is an important species for industrial bioprocesses as it is easy to genetically engineer and is relatively resistant to harsh conditions. Herein, we demonstrate that the possession of the vacuolar membrane H^+^-PPase can broaden this resistance for several such conditions. In this study, it has been demonstrated that by the addition of a vacuolar membrane H^+^-PPase, there is a positive impact on growth rates, as well as a reduction in the lag phase when the strains are grown at a low pH with acetic acid as an inhibitor.

Additionally, the anaerobic xylose metabolism of the *S. cerevisiae* strains used herein are compromised due to a redox burden together with an inadequate ATP formation flux. On top of that, high expression levels of the biosensors were required to have a significant fluorescent signal distinguishable from auto-fluorescens. Therefore, it was no surprise that the biosensors implemented in these strains demanded even more energy. However, the vacuolar membrane H^+^-PPase expression with pHluorin unexpectedly lifted this burden, thus restoring anaerobic xylose metabolism. Additionally, there is a 20% reduction in the fermentation duration when the vacuolar membrane H^+^-PPase is expressed with pHluorin compared to the parent strain used. The mechanistic function of this H^+^-PPase is more complex than was anticipated, and merits further dedicated investigation.

Overall, this study identifies a new potential avenue for improving the production of high-value compounds that are limited by ATP levels. Implementing the H^+^-PPase of *A. thaliana* in yeast may provide several advantages due to the bidirectional and the substrate-independent nature of the proton pumps. This may contribute to a more flexible way for the cell to maintain its pH_i_ homeostasis, and thus, secure and maintain metabolic performance under industrially challenging conditions, which is of future interest.

## Figures and Tables

**Figure 1 microorganisms-12-00625-f001:**
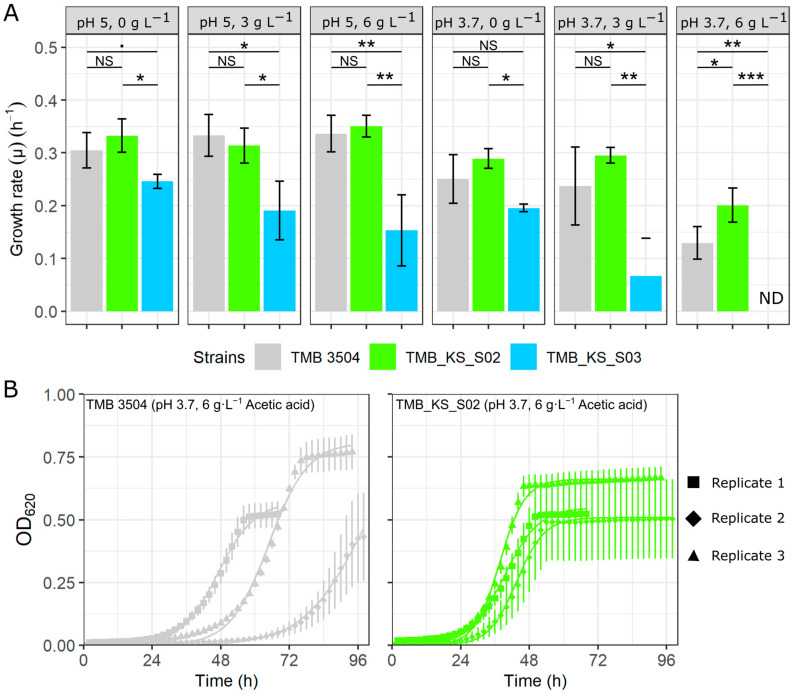
Growth characteristics of the control (TMB 3504), vacuolar membrane H^+^-PPase (TMB_KS_S02), and cell membrane H^+^-PPase (TMB_KS_S03) strains grown in a Verduyn mineral media with 20 g·L^−1^ glucose and 0, 3, and 6 g·L^−1^ of acetic acid at an initial pH of 5 and 3.7 in microtiter plates. (**A**) Growth rates of the three strains over three biological replicates at different concentrations of acetic acid in minimal media at two different pHs (OD_620_ = optical density at 620 nm). The error bars represent the standard deviation between the biological replicates. [NS represents a *p*-value greater than 0.1, (.) represents a *p*-value between 0.05 and 0.1, (*) represents a *p*-value between 0.01 and 0.05, (**) represents a *p*-value between 0.001 and 0.01, (***) represents a *p*-value between 0 and 0.001]. (**B**) The growth profiles of the parent strain (TMB 3504) (Grey) and the vacuolar membrane H^+^-PPase strain (TMB_KS_S02) (Green) under the most stressful condition tested (pH 3.7, 6 g·L^−1^ acetic acid). The square, diamond and the triangle shapes are biological replicates with three technical replicates (three individual wells inoculated from separate colonies obtained from a single clone) represented as the standard deviations for each biological replicate. The solid lines are the logistic models fitted through the technical replicates for each biological replicate. ND = not determined.

**Figure 2 microorganisms-12-00625-f002:**
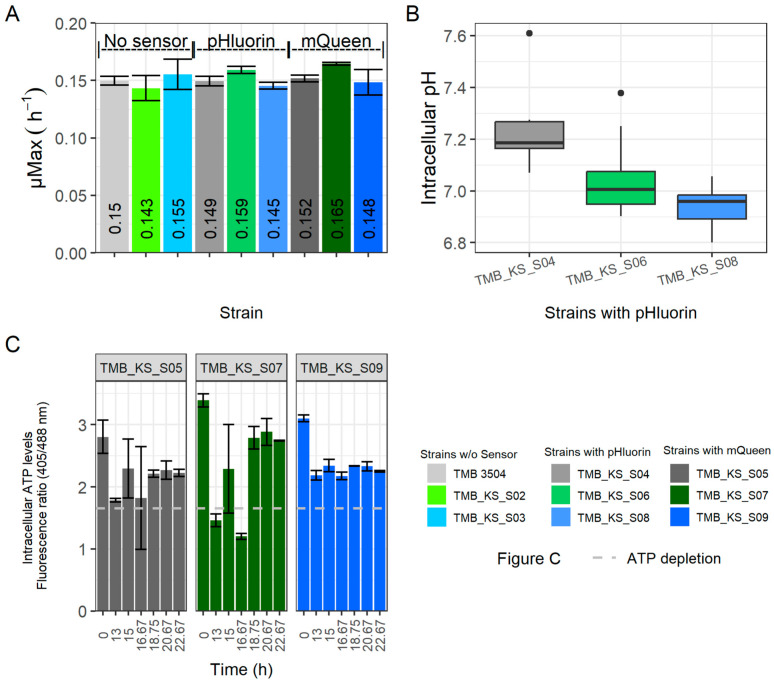
Observed factors for anaerobic glucose fermentations. (**A**) Growth rates of the various strains [error bars represent the standard deviations between biological duplicates], (**B**) pH_i_ determined using pHluorin biosensor [the box represents the quartiles of all the pH readings obtained across the biological duplicates and the outliers are represented as dots]. (**C**) Intracellular ATP concentration detected with QUEEN-2m. pHluorin and QUEEN-2m measurements are made across two biological duplicates (represented as standard deviations) at distinct time points during glucose fermentation. The dashed grey line in (**C**) represents the ATP depletion condition established by resuspending the cells in mineral media with 0.5 g·L^−1^ of 2-deoxy-D-glucose for 1 h. TMB_3504 is the parent strain, and TMB_KS_S02 and TMB_KS_S03 are its derivatives with the proton pump targeted to the vacuolar and cytosolic membrane, respectively. TMB_KS_S04, TMB_KS_S06 and TMB_KS_S08 are the derivatives of TMB_3504, TMB_KS_S02 and TMB_KS_S03, respectively, with the pHluorin biosensor. TMB_KS_S05, TMB_KS_S07 and TMB_KS_S09 are the derivatives of TMB_3504, TMB_KS_S02 and TMB_KS_S03, respectively, with the QUEEN-2m biosensor.

**Figure 3 microorganisms-12-00625-f003:**
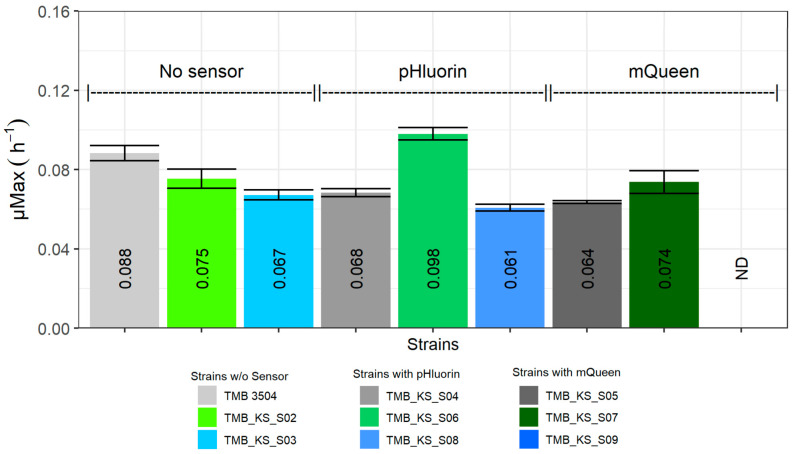
Growth rates of the various strains during anaerobic xylose fermentations [error bars represent the standard deviations between biological duplicates]. ND = not determined.

**Figure 4 microorganisms-12-00625-f004:**
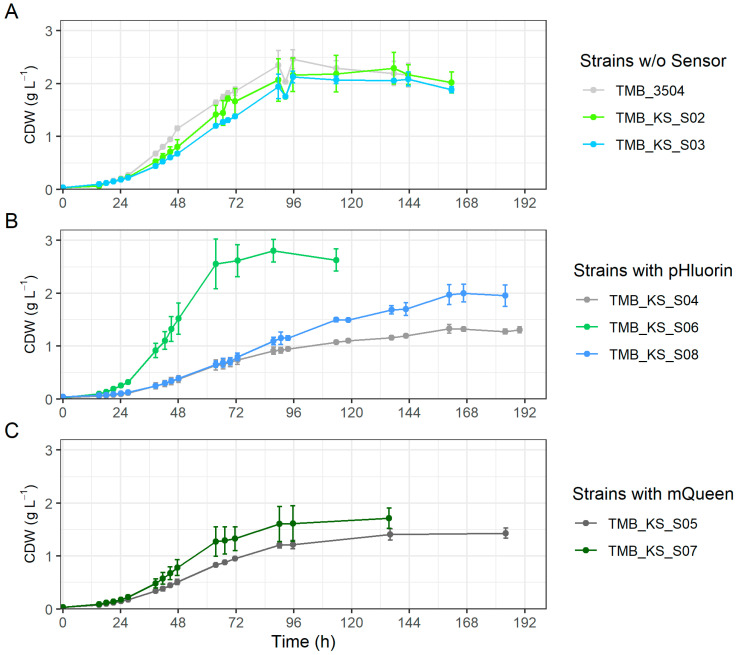
Time series of cell dry weights (CDW) for anaerobic fermentations on xylose 50 g·L^−1^ in bioreactors. (**A**) The parent strain, the vacuolar, and the cytosolic strain. (**B**) Derivative strains of (**A**) with pHluorin. (**C**) Derivative strains of (**A**) with QUEEN-2m. The metabolic profiles for these fermentations are shown in Appendix A. The error bars are the standard deviations obtained from biological duplicates.

**Figure 5 microorganisms-12-00625-f005:**
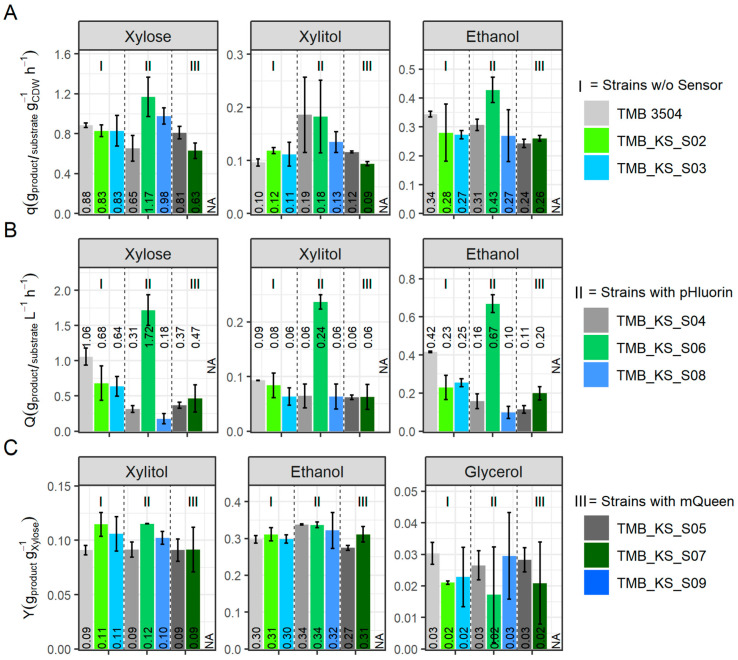
Specific productivity (q), volumetric productivity (Q), and yield (Y) of the various strains grown on 50 g·L^−1^ xylose in bioreactors. [Standard deviations between replicates are represented as error bars]. (**A**) The q_xylose_, q_xylitol_, and q_ethanol_ calculated during logarithmic growth. (**B**) The Q_xylose_, Q_xylitol_, and Q_ethanol_ calculated during logarithmic growth. (**C**) The Y_xylose_, Y_xylitol_, and Y_ethanol_ over the entire fermentation period. TMB 3504 is the parent strain, and TMB_KS_S02 and TMB_KS_S03 are its derivatives with the proton pump targeted to the vacuolar and cytosolic membrane, respectively. TMB_KS_S04, TMB_KS_S06 and TMB_KS_S08 are the derivatives of TMB_3504, TMB_KS_S02 and TMB_KS_S03, respectively, with the pHluorin biosensor. TMB_KS_S05, TMB_KS_S07, and TMB_KS_S09 are the derivatives of TMB_3504, TMB_KS_S02, and TMB_KS_S03, respectively, with the QUEEN-2m biosensor. NA = not applicable.

**Figure 6 microorganisms-12-00625-f006:**
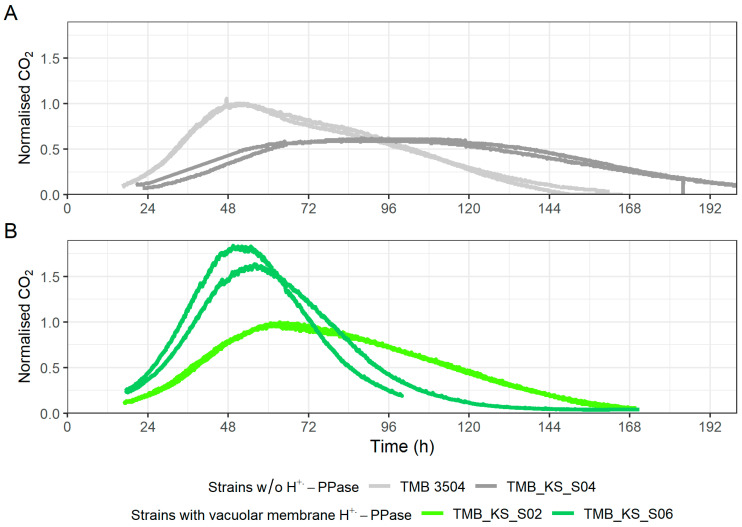
Time course of the carbon dioxide (CO_2_) production profiles for each of the biological replicates for various fermentations carried out in bioreactors (measured at either 5 or 10 s intervals (Section 2.6)). The first ~18 h of the fermentation was carried out without sparging and hence removed from the image. (**A**) CO_2_ production profile for the parent strain (TMB 3504) and the parent strain with pHluorin (TMB_KS_S04) normalized to the parent strain (TMB 3504). (**B**) CO_2_ production profile for the vacuolar membrane H^+^-PPase strain (TMB_KS_S02) and the vacuolar membrane H^+^-PPase strain with pHluorin (TMB_KS_S06) normalized to the vacuolar membrane H^+^-PPase strain (TMB_KS_S02). The CO_2_ production profiles for the strains with pHluorin are normalized to its ancestor strain.

**Figure 7 microorganisms-12-00625-f007:**
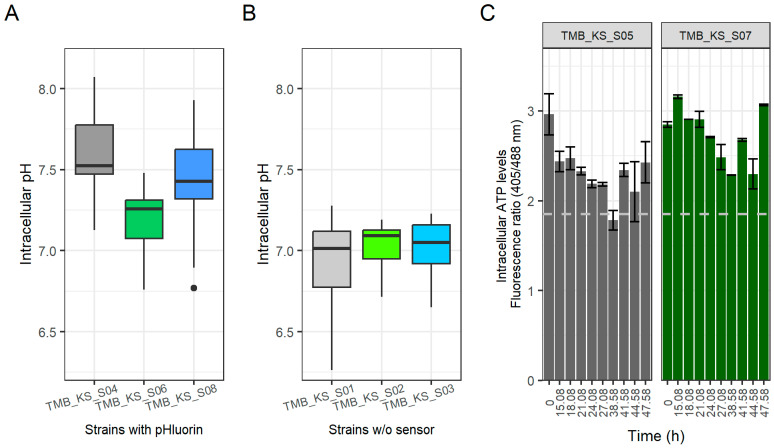
(**A**) Intracellular pH determined using pHluorin [the box represents the quartiles of all the pH readings obtained across the biological duplicates and the outliers are represented as dots]. (**B**) Intracellular pH determined using pHrodo^®^ green [the box represents the quartiles of all the pH readings obtained across the biological duplicates]. (**C**) Intracellular ATP concentration detected with QUEEN-2m. pHluorin and QUEEN-2m measurements are made across two biological duplicates (represented as standard deviations) at distinct time points during glucose fermentation. The dashed grey line defines the ATP depletion condition. (TMB_3504 is the parent strain, and TMB_KS_S02 and TMB_KS_S03 are its derivatives with the proton pump targeted to the vacuolar and cytosolic membrane, respectively. TMB_KS_S04, TMB_KS_S06 and TMB_KS_S08 are the derivatives of TMB_3504, TMB_KS_S02, and TMB_KS_S03, respectively, with the pHluorin biosensor. TMB_KS_S05 and TMB_KS_S07 are the derivatives of TMB_3504 and TMB_KS_S02, respectively, with the QUEEN-2m biosensor).

**Table 1 microorganisms-12-00625-t001:** List of the nine *S. cerevisiae* strains used in this study. ((Tc) is the signal peptide from *Trypanosoma cruzi*; (Suc2) is the signal peptide (first 25 amino acids) of invertase as used in Drake et al. [34]).

Strain Designation	Strain Name	Relevant Genotype	References
TMB 3504	Parent Strain	CEN.PK 2-1C; *MATa*; *ura3-52*; Δ*gre3*; *his3*::*HIS3 PGK1p-XKS1-PGK1t*; *TAL1*::*PGK1p-TAL1-PGK1t*; *TKL1*::*PGK1p-TKL1-PGK1t*; *RKI1*::*PGK1p-RKI1-PGK1t*; *RPE1*::*PGK1p-RPE1-PGK1t*; *ura3*::*YIpRC5p*	[43]
TMB_KS_S02	Vacuolar membrane H^+^-PPase strain	TMB 3504; XI-3::*TEF1p-*(Tc)*AVP1-CYC7*t	This study
TMB_KS_S03	Cell membrane H^+^-PPase strain	TMB 3504; XI-3::*TEF1p-*(Suc2)*AVP1-CYC7*t	This study
TMB_KS_S04	Parent strain with pHluorin	TMB 3504; X-4::*GPD1p-pHluorin-CYC7*t	This study
TMB_KS_S05	Parent strain with mQueen	TMB 3504; X-4::*GPD1p-mQueen2m-CYC7*t	This study
TMB_KS_S06	Vacuolar membrane H^+^-PPase strain with pHluorin	TMB KS S02; X-4::*GPD1p-pHluorin-CYC7*t	This study
TMB_KS_S07	Vacuolar membrane H^+^-PPase strain with mQueen	TMB KS S02; X-4::*GPD1p-mQueen2m-CYC7*t	This study
TMB_KS_S08	Cell membrane H^+^-PPase strain with pHluorin	TMB KS S03; X-4::*GPD1p-pHluorin-CYC7*t	This study
TMB_KS_S09	Cell membrane H^+^-PPase strain with mQueen	TMB KS S03; X-4::*GPD1p-mQueen2m-CYC7*t	This study

**Table 2 microorganisms-12-00625-t002:** Plasmids used in this study. ((Tc) is the signal peptide from *Trypanosoma cruzi*; (Suc2) is the signal peptide (first 25 amino acids) of invertase as used in Drake et al. [34]).

Name	Relevant Genotype	Source
pRSET-QUE2m	ColE1; AmpR; *T7p-QUEEN-2m-T7*t	[40] [Addgene; #129350]
pUC57-VP-Suc2PSP-AVP1	AmpR; *M13p-TEF1p-*(Tc)(Suc2)*AVP1-CYC7*t	This study
pYES-pACT1-pHluorin	AmpR; URA3; *ACT1p-pHluorin-CYC1*t	[39]
p426-GPDp	URA3; AmpR	[45]
pTMB_KS_036	AmpR; URA3; *GPD1p-pHluorin-CYC7*t;	This Study
pTMB_KS_038	AmpR; URA3; *GPD1p-Queen-2m-CYC7*t;	This Study
pCFB2312	KanR; pTEF1p-Cas9-CYC1t	[46]
pCFB3035	gRNA_X-4; natMX	[46]
pCFB3042	X-4 MarkerFree backbone; Geneticin	[46]
pCFB2904	gRNA_XI-3; natMX	[46]
pCFB3045	XI-3 MarkerFree backbone; Geneticin	[46]
pTMB_KS_040	pCFB3042; *GPD1p-pHluorin-CYC7*t	This Study
pTMB_KS_041	pCFB3042; *GPD1p-Queen-2m-CYC7*t	This Study
pTMB_KS_042	AmpR; *M13p-TEF1p-*(Tc)*AVP1-CYC7*t	This Study
pTMB_KS_043	AmpR; *M13p-TEF1p-*(Suc2)*AVP1-CYC7*t	This Study
pTMB_KS_044	pCFB3045; *TEF1p-*(Tc)*AVP1-CYC7*t	This Study
pTMB_KS_045	pCFB3045; *TEF1p-*(Suc2)*AVP1-CYC7*t	This Study

## Data Availability

Data are contained within the article or Appendix A. Any additional datasets are available from the correspondence author on reasonable request.

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
