# Peer review of "Evaluation of Pyrophosphate-Driven Proton Pumps in Saccharomyces cerevisiae under Stress Conditions"

_microorganisms, 2024, doi:10.3390/microorganisms12030625_

Round 1

Reviewer 1 Report

Comments and Suggestions for Authors

This is a clear and well written paper describing the effects of expression of a plant-derived pyrophosphate-driven proton pump in either cell or vacuolar membrane of S. cerevisiae, and its effects on growth under various conditions. The experiments and results are well described and are of interest to others working in this field.

Minor comments and questions:

Table 1, row beginning 'TMB_KS_S07': missing word after 'with' in column 2?

Section 2.6, second paragraph - what detector was used for HPLC?

Figure 1A: apologies if I am misunderstanding, but are these box and whisker plots showing the median and quartile for the measurements in each case, being three technical replicates of three biological replicates? But from looking at the fitting procedure in Supplementary Figures S2 and S3, it seems that you should only derive three growth rates for each condition? In which case, what do the values in these boxes and whiskers represent? Also, would it be possible to provide an indication in the figure, caption or text as to which differences are statistically significant and at what level of significance?

The same comment applies to Figure 2B and Supplementary Figure S4.

In Figure 1B, as well as some panels in S2 and S3, the standard deviations of the technical replicates seem very high for an OD measurement, which I would have expected to be reasonably simple and precise. Or have I misunderstood? Could it be clarified in section 2.4, are the technical replicates multiple wells and the biological replicates separate clones? or are the technical replicates multiple measurements of a single well?

In Figure 2A and 2C, what do the error bars represent? Can this be stated in the caption? Also, is it possible to indicate which differences are statistically significant? Statistical significance should also be mentioned in the text, eg in line 372 which mentions 'slight acidification'.

Figure 3: essentially similar comments to those above.

Text following Figure 3: these are quite remarkable and unexpected results, or so it seems to me. Could it be clarified whether the biological replicates were from multiple independent cloning events, preferably multiple different transformation experiments? If the replicates were all from the same clone, one might be inclined to wonder whether some other genetic change or rearrangement may have occurred in this clone, possibly relating to the engineered xylose utilisation pathway. The apparent synergy between the vacuolar H+-PPase and pHlourin strikes me as so surprising that I think it would be of interest to repeat the entire construction process and check that the same phenomenon is observed (but if this was not done, I don't propose that it needs to be done for this paper - just that this point should be clarified). I am not sure that I find the proposed mechanism in lines 459-462, 490-491 and 499-501 altogether convincing, and perhaps the authors would like to expand on this and clarify it; however, I would have no problem with presenting this and allowing the reader to form their own opinion.

Figure 4: state in caption what error bars indicate?

Lines 549-552, again please mention statistical significance of these results?

Lines 566-567, same comment.

Line 581-583 seems to assume the correctness of the hypothesis stated above. I would prefer to see demonstrations of the same effect with burden caused by expression of other heterologous proteins before drawing any such conclusion. Perhaps qualify this statement a bit? Lines 617-619 are more circumspect.

Line 584 'on fermentation of lignocellulosic biomass'?

Comments on the Quality of English Language

The English language usage is generally excellent. There are a few very minor issues which can be corrected during editing, for example:

line 33 'polyketides'

line 35 'pumps'

line 38 'have'

line 49 'conditions'

line 70 'to overexpress'

line 77 'which is'

line 95 'severity'

line 149 'heat shock'

line 157 'was done'

lines 160, 162, 164, 177, 222 and elsewhere: note that 'media' is plural and in most of these places, 'medium' would be more appropriate.

line 183 'inoculation ... was' or 'inoculations ... were'

line 254 'subjected to'

line 288 'compensate for'

line 345 delete 'that'

line 409 'vacuolar'

lines 543-544 'mechanisms ... remain' or 'mechanism ... remains'

line 611 'independently of whether'?

Reviewer 2 Report

Comments and Suggestions for Authors

S. cerevisiae is an important species for many biological application. I feel that the manuscript  offer an interesting and new topic and it should be published. English is o.k..

I signed only some minor gramathic problems (unnecessary spaces, typos, etc.). Authors should check the entire text carefully. Literature: DOI is not necessary and if you want to use it, I think that better is in a from: https//xxx and no DOI:xxx.

Author Response

Review 2

S. cerevisiae is an important species for many biological application. I feel that the manuscript  offer an interesting and new topic and it should be published. English is o.k..

I signed only some minor gramathic problems (unnecessary spaces, typos, etc.). Authors should check the entire text carefully. Literature: DOI is not necessary and if you want to use it, I think that better is in a from: https//xxx and no DOI:xxx.

We thank reviewer 2 for her/his observations. Because the other reviewers have also spotted many little grammatical errors and typos and listed them, we focused on these. We have followed the reference style provided by the journal which calls for the DOI numbers so we cannot put it in the https from.

Reviewer 3 Report

Comments and Suggestions for Authors

The manuscript submitted to “Microorganisms” an MDPI journal entitled: “Evaluation of pyrophosphate-driven proton pumps in Saccharomyces cerevisiae under stress conditions” which investigated in Saccharomyces cerevisiae, pH homeostasis is reliant on ATP due to the use of ATP- driven proton pumps (H+-ATPase) which constitute a major drain on the cellular ATP supply. Here, an exogenous pyrophosphate-driven proton pump (H+-PPase) from Arabidopsis thaliana, which uses inorganic pyrophosphate (PPi) rather than ATP, was evaluated for its effect on the ATP burden during cultivation conditions known to affect the intracellular pH, the following comments should be followed:

-          Generally the idea of the paper is not obvious and the manuscript is corrupted as well as too many non-significant details are mentioned also the significant points are missed in the paper, so extreme editing in whole the manuscript is urgent in order to make sense and compatibility with all chapters of the paper.  

-          All names of fungi and plants (genus and species) should be italic in whole part of the manuscript as well as in references section.

-          All genes’ names should be corrected in whole manuscript even the figures as the first 3 letters are small and italic and the fourth letter is capital and non-italic.

-          The numbers should be subscript in whole manuscript for example CO2.

-          Key words: harmonize the key words with the words of abstract.

-          All abbreviated words firstly should be written with complete word then abbreviate in whole manuscript.

-          The titles, subtitles or the beginning of the paragraph should contain whole words then abbreviate.

-          Add source and ID number of used strains, and data about strains are missed as: year of isolation, locality of isolation as well as isolation and reconfirmation of identification should be performed before using it and mention that in materials and methods section.

-          Abstract: is not clear towards the main points so should be re-written to be more comprehensive about the highlighted results in order to sounds better.

-          Lines 84-95: should be rewritten in more condensed form and remove the references here as this part is your aim of the work.

-          Materials and Methods:

·         Add the references in each step because some of them are missed.

·         Mention the number of used yeast strains.

·         Explanation of the tables should be written above each one as well as figures in whole the manuscript.

·         Remove the references from the title of the tables in whole the manuscript.

·         Mention the ID number of E. coli and the source.

·         Mention the source of all used media.

-          Results

·         Improve the resolution of all figures.

·         Many details were written in scattered form, please rearrange the data in this section.

-          Discussion:

·         Was poorly written and should make comparison with each result with recent relevant references with your own explanation of the difference or similarity.

-          Conclusion: is very short, please rewrite as well as remove the references here and mention your own obtained data and the conclusion.

-          References:

·         Should be updated till 2023.

·         More relevant references about the aim of the study should be added.

·         Write the all authors, names not et al. in all references.

·         Unify the style of references writing according to the author guide of the journal.

-          Section of abbreviation at the end of the manuscript should be added.

-          Finally, what is the impact of the obtained results practically and what are their applications?

Comments on the Quality of English Language

 Minor editing of English language required

Round 2

Reviewer 3 Report

Comments and Suggestions for Authors

Really, a lot of things even all required corrections have not performed, please follow these comments and correct with highlight the modified corrections within the manuscript as followed:

The manuscript submitted to “Microorganisms” an MDPI journal entitled: “Evaluation of pyrophosphate-driven proton pumps in Saccharomyces cerevisiae under stress conditions” which investigated in Saccharomyces cerevisiae, pH homeostasis is reliant on ATP due to the use of ATP- driven proton pumps (H+-ATPase) which constitute a major drain on the cellular ATP supply. Here, an exogenous pyrophosphate-driven proton pump (H+-PPase) from Arabidopsis thaliana, which uses inorganic pyrophosphate (PPi) rather than ATP, was evaluated for its effect on the ATP burden during cultivation conditions known to affect the intracellular pH, the following comments should be followed:

-          Generally the idea of the paper is not obvious and the manuscript is corrupted as well as too many non-significant details are mentioned also the significant points are missed in the paper, so extreme editing in whole the manuscript is urgent in order to make sense and compatibility with all chapters of the paper.  

-          All names of fungi and plants (genus and species) should be italic in whole part of the manuscript as well as in references section.

-          All genes’ names should be corrected in whole manuscript even the figures as the first 3 letters are small and italic and the fourth letter is capital and non-italic.

-          The numbers should be subscript in whole manuscript for example CO2.

-          Key words: harmonize the key words with the words of abstract.

-          All abbreviated words firstly should be written with complete word then abbreviate in whole manuscript.

-          The titles, subtitles or the beginning of the paragraph should contain whole words then abbreviate.

-          Add source and ID number of used strains, and data about strains are missed as: year of isolation, locality of isolation as well as isolation and reconfirmation of identification should be performed before using it and mention that in materials and methods section.

-          Abstract: is not clear towards the main points so should be re-written to be more comprehensive about the highlighted results in order to sounds better.

-          Lines 84-95: should be rewritten in more condensed form and remove the references here as this part is your aim of the work.

-          Materials and Methods:

·         Add the references in each step because some of them are missed.

·         Mention the number of used yeast strains.

·         Explanation of the tables should be written above each one as well as figures in whole the manuscript.

·         Remove the references from the title of the tables in whole the manuscript.

·         Mention the ID number of E. coli and the source.

·         Mention the source of all used media.

-          Results

·         Improve the resolution of all figures.

·         Many details were written in scattered form, please rearrange the data in this section.

-          Discussion:

·         Was poorly written and should make comparison with each result with recent relevant references with your own explanation of the difference or similarity.

-          Conclusion: is very short, please rewrite as well as remove the references here and mention your own obtained data and the conclusion.

-          References:

·         Should be updated till 2023.

·         More relevant references about the aim of the study should be added.

·         Write the all authors, names not et al. in all references.

·         Unify the style of references writing according to the author guide of the journal.

-          Section of abbreviation at the end of the manuscript should be added.

-          Finally, what is the impact of the obtained results practically and what are their applications?

Comments on the Quality of English Language

Minor editing of English language required

Author Response

Authors have responded all the comments.